# Unsupervised Text Generation by Learning from Search

**Jingjing Li[1], Zichao Li[2], Lili Mou[3], Xin Jiang[2], Michael R. Lyu[1], Irwin King[1]**
[1]The Chinese University of Hong Kong   [2]Huawei Noah's Ark Lab
[3]University of Alberta; Alberta Machine Intelligence Institute (Amii)
{lijj,lyu,king}@cse.cuhk.edu.hk
{li.zichao,jiang.xin}@huawei.com
doublepower.mou@gmail.com

## Abstract

In this work, we propose TGLS, a novel framework for unsupervised Text Generation by Learning from Search. We start by applying a strong search algorithm (in particular, simulated annealing) towards a heuristically defined objective that (roughly) estimates the quality of sentences. Then, a conditional generative model learns from the search results, and meanwhile smooth out the noise of search. The alternation between search and learning can be repeated for performance bootstrapping. We demonstrate the effectiveness of TGLS on two real-world natural language generation tasks, unsupervised paraphrasing and text formalization. Our model significantly outperforms unsupervised baseline methods in both tasks. Especially, it achieves comparable performance to strong supervised methods for paraphrase generation.[1]

## 1   Introduction

Text generation refers to a wide range of tasks involving generating natural language, including machine translation [19, 20, 18], sentence simplification [29, 41], and text summarization [5, 1]. Recent success of neural text generation relies heavily on large parallel data for training, which may not be available in real-world natural language processing (NLP) applications. In this work, we consider unsupervised text generation, where no parallel data is available. This setting is more challenging, and has significant potential in both scientific research (e.g., low-resource language processing) and industrial applications (e.g., cold start for a new NLP application).

Early work tackles unsupervised text generation by rules or templates [47, 27]. While such approaches do not require parallel corpora, the generated sentences are highly subject to the rules, and hence lack the flexibility of natural language. Other work constructs pseudo-parallel data, which is only feasible for certain tasks like unsupervised machine translation [19].

Recently, researchers have developed search-based techniques for unsupervised text generation [28, 17, 37, 23], where a heuristically defined scoring function evaluates the quality of a sentence, involving language fluency, semantic compliance, and other task-specific aspects. Then, the algorithm performs word-level edits (such as word deletion, insertion, and replacement) to search towards a (possibly local) optimum of the scoring function. With a reasonably designed scoring function, such approaches are shown to be effective in a variety of applications like paraphrase generation [28, 23], sentence summarization [37], and text simplification [17].

However, the above search-based approach has two major drawbacks: 1) The inference efficiency is low. To obtain an output sentence, the search algorithm would perform a few hundred steps of local

edits and re-evaluations. This could be considerably slower than an autoregressive decoder, which generates words sequentially. 2) The search could yield noisy results, since the scoring function is defined heuristically and the search is conducted locally in a discrete sentence space.

To this end, we propose a new framework for unsupervised TEXT GENERATION by LEARNING FROM SEARCH (TGLS), which contains a strong search module that explores the sentence space, as well as a learning module that learns from the search results. For the search module, we adopt the simulated annealing (SA) algorithm. At each step, SA proposes a local edit by a neural network, and then either accepts or rejects the proposal based on a heuristically defined scoring function. For learning, we employ two methods to train a conditional generative model, word-level cross-entropy loss and the sequence-level max-margin loss. Within TGLS, the search and learning can be boosted by each other in an iterative fashion. That is, the search results serve as the pseudo-reference for training the conditional generator, which in turn benefits SA search by serving as a more meaningful initial state. As for implementation, TGLS involves two pretrained language models: 1) the uni-directional GPT2 [33], which is suitable for likelihood-based fluency evaluation and conditional generation; and 2) the bi-directional RoBERTa [24], which is better at semantic evaluation and contextual word-level prediction.

The main contributions of our paper include: 1) We propose TGLS, a principled framework for unsupervised text generation; TGLS can be applied to different tasks if the output resembles the input and can be roughly estimated by a heuristically defined scoring function. 2) We successfully incorporate large-scale pretrained language models into our TGLS framework. 3) We conducted experiments on two different tasks: paraphrasing and text formalization. In both experiments, TGLS significantly outperforms unsupervised baseline methods. Moreover, TGLS achieves comparable performance to recent supervised models [7] in the paraphrasing task. 4) For text formalization (an example of text style transfer), we are also the first to design a search-based method, and further extend it into the proposed TGLS framework.

## 2 Approach

Our TGLS framework involves two stages of search and learning. In the first stage, we perform simulated annealing (SA) search [23] and treat the obtained output sentences as pseudo-references. Then, we train an autoregressive GPT2 as the text generator [33] by word-level cross-entropy (CE) supervised learning, which enables our model to learn quickly. In the second stage, the GPT2 performs beam search and the output is taken as the initial state of the SA algorithm again for iterative performance improvement. Later, we perform max-margin (MM) learning to better distinguish between higher-scored sentences and other high-probability but sub-optimal sentences. Figure 1 provides an overview of the two stages of search and learning in TGLS.

### 2.1 Simulated Annealing Search

The search-based text generation [28, 23] relies on a heuristic-based objective function $s(\mathrm{y}|\mathrm{x})$ that (roughly) evaluates the quality of an output sequence $\mathrm{y}$ given the input $\mathrm{x}$ (usually, one or a few sentences). Typically, the objective involves language modeling fluency $s_{\mathrm{lm}}(\mathrm{x})$, semantic compliance $s_{\mathrm{semantic}}(\mathrm{x}, \mathrm{y})$, and other task-specific scorers $s_{\mathrm{task}}(\mathrm{y}, \cdot)$. These individual scorers are combined by the product of experts [13]:

$$s(\mathrm{y}|\mathrm{x}) = s_{\mathrm{lm}}(\mathrm{y}) \cdot s_{\mathrm{semantic}}(\mathrm{x}, \mathrm{y}) \cdot s_{\mathrm{task}}(\mathrm{y}, \cdot). \tag{1}$$

We adopt simulated annealing (SA) [16, 23], which performs local stochastic search to maximize the objective. Concretely, SA starts from an initial candidate output sentence $\mathrm{y}^{(0)}$, which is set to the input $\mathrm{x}$ in our first-stage SA. For the second stage, it will be the output of our GPT2 model.

At a search step $t$, SA iteratively proposes a new candidate $\mathrm{y}'$ by local edits of $\mathrm{y}^{(t)}$, namely, word insertion, deletion, and replacement. The proposal $\mathrm{y}'$ is accepted with probability $p(\mathrm{accept}|\mathrm{y}', \mathrm{y}^{(t)}, \mathrm{x}, T) = \min\{1, \exp(\frac{s(\mathrm{y}'|\mathrm{x}) - s(\mathrm{y}^{(t)}|\mathrm{x})}{T})\}$. Then, $\mathrm{y}^{(t+1)} = \mathrm{y}'$ if $\mathrm{y}'$ is accepted, or otherwise, $\mathrm{y}^{(t+1)} = \mathrm{y}^{(t)}$. In SA, $T$ is a temperature controlling how greedy the search algorithm is. Usually, $T$ is high at the beginning of search so as to be more explorative, and then $T$ is cooled down to achieve a better (local) optimum. Although we follow the generic SA framework of text generation as in [23], the objective function and proposal are largely redesigned, detailed below.

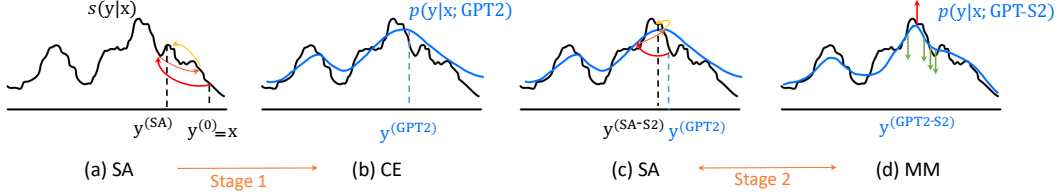

Figure 1: Overview of TGLS. (a) First-stage search by simulated anealing (SA). (b) First-stage learning by cross-entropy (CE) loss. (c) Second-stage search by SA. (d) Second-stage learning by max-margin (MM) loss. The horizontal axis represents the sentence space.

**Fluency scorer ($s_{lm}$).** The fluency of a sentence can oftentimes be approximated by a language model's predicted probability. Previous search-based work uses recurrent neural networks for fluency evaluation [28, 23]. In our work, we use the large-scale pretrained GPT2 model [33]. For an output $y = y_1 \cdots y_n$, the language fluency scorer is the joint likelihood of $y$, given by $s_{lm}(y) = (\prod_{i=1}^{n} p(y_i|y_1, \cdots, y_{i-1}))^{\alpha}$, where $\alpha$ is a hyperparameter balancing $s_{lm}$ with other scorers in (1). In fact, we use the vocabulary of GPT2 with bype-pair encoding (BPE), and $y_i$ here is a token after BPE segmentation. Our GPT2 is fine-tuned with non-parallel in-domain corpora to learn the specificity of a task.

**Semantic scorer ($s_{semantic}$).** In this part, we extend the semantic scorers in [23] with a RoBERTa [24]. Fine-tuning details are presented in Appendix A. Compared with autoregressive GPT2 used for fluency evaluation, RoBERTa is pretrained by masked language modeling, and is better at feature representation. Let $x = (x_1, \cdots, x_m)$ be a sentence. RoBERTa computes a contexualized representation of a word in the sentence as RoBERTa($x_i$, x).

A word-level semantic scorer evaluates how much keyword information (detected by Rake [36]) is preserved, given by the least matched keyword of x:

$$s_{\text{word}}(y, x) = \min_{k \in \text{keyword}(x)} \max_{y_i \in y} \text{RoBERTa}(k, x)^{\top} \text{RoBERTa}(y_i, y). \tag{2}$$

A sentence-level semantic scorer evaluates the cosine similarity of two sentence vectors $s_{\text{sent}}(y, x) = \frac{y^{\top} x}{\|y\|\|x\|}$, where the sentence vector is given by the RoBERTa feature of the padded token [BOS] at the beginning end of a sentence, i.e., $x = \text{RoBERTa}([\text{BOS}], x)$ and $y$ is computed analogously.

Finally, the semantic scorer is the product of both word- and sentence-level scores as

$$s_{\text{semantic}}(y, x) = s_{\text{word}}(y, x)^{\beta} \cdot s_{\text{sent}}(y, x)^{\gamma}, \tag{3}$$

where $\beta$ and $\gamma$ are weighting hyperparameters.

**Task-specific scorers.** We apply TGLS to two tasks: paraphrasing and text formalization.

For paraphrasing, the goal is to generate a semantically similar but lexically different sentence. Previous work [23] uses the BLEU score to penalize the $n$-gram overlapping between the output and input: $s_{\text{paraphrase}}(y, x) = (1 - \text{BLEU}(y, x))^{\delta}$, which is also adopted in our work. Here, $\delta$ is a weighting hyperparameter for the task-specific scorer.

For text formalization, the goal is to transform an informal sentence to the formal style [35], which is an example of text style transfer. We follow the setting of most text style-transfer work [14], where we assume the style labels are available, but no parallel supervision is given. We train a classifier that predicts the probability of the style, also based on the RoBERTa features. Then, the task-specific scorer becomes $s_{\text{formality}}(y) = p(\text{formal} | \text{RoBERTa}([\text{BOS}], y))^{\delta}$, where $\delta$ is the weighting hyparaparameter for this task.

**Proposal of local edits.** At a step $t$ of SA search, a new candidate $y'$ is proposed from $y^{(t)}$ by local editing. SA randomly picks a position to edit, as well as one of the following operators: Replace, Insert, and Delete.

For Replace, the model suggests a candidate word at $x_i$ based on the posterior distribution induced by $s(y|x)$. For efficiency concerns, previous work [28, 23] evaluates top-$K$ candidate words, suggested by a forward and backward language model. In our work, we adopt RoBERTa to evaluate the

posterior probability of a word, where the word embedding layer of RoBERTa at this slot is randomly masked. The `Insert` edit also suggests a word from the posterior, predicting a word given the newly added [MASK] token and the context. This complies with RoBERTa's pretraining criteria of masked language modeling and is able to suggest high-quality candidate words. The `Delete` operator simply removes the word at a chosen position.

In text formalization, we also have rule-based local edits (e.g., "*we are*" substituting "*we're*") which are retrieved from PPDB [30]. Previous sequence-to-sequence approaches on this task adopt manually designed rules as a preprocessing step [35] or as additional input concatenated with the raw sentence [46]. Our unsupervised TGLS, on the other hand, can easily make use of the off-the-shelf resources, and can easily filter out the noise by rejecting bad candidates.

In short, the SA search component in our TGLS mainly follows [23], but we re-design the scoring functions and the proposals. The main focus of this paper is to couple search and learning, especially the methods of training a machine learning model that learns from the search results, as follows.

## 2.2 Word-Level Cross-Entropy (CE) Learning

As mentioned in Section 1, the local search algorithm is computationally inefficient during inference time, because it requires a few hundred steps of edits and re-evaluations for each sample.

Our intuition is to train a conditional generative model, GPT2, based on SA's search results. Specifically, we concatenate an input x and SA's searched sequence $y^{(SA)}$ with a special separating token [SEP] in between, and train GPT2 with losses on the y-part. Therefore, the GPT2 would be able to generate an output sequence directly from $p(y|x)$ in an autoregressive way.

Given a source sequence x, the objective is the word-by-word cross-entropy (CE) loss, given by

$$J_{CE} = - \sum_{i=1}^{N} \sum_{v \in \mathcal{V}} y_{i,v}^{(SA)} \log p_{i,v}^{(GPT2)}, \tag{4}$$

where $y_{i,v}^{(SA)}$ is a binary value, indicating whether the $i$th word is $v$ or not in the SA's output for this data sample, $N$ is the length of y, and $p_{i,v}^{(GPT2)} = \Pr\left[y_i = v \mid y_{<i}^{(SA)}, x\right]$, which is predicted by the GPT2.

The word-level CE learning in TGLS adopts standard teacher-forcing technique with SA's output being the pseudo-reference, i.e., during training, the GPT2 model learns the probability $p_{i,v}^{(GPT2)}$ at step $i$, assuming all previous words are correctly predicted as $y_{<i}^{(SA)}$. Thus, word-by-word CE trains all predictions in the sequence simultaneously, and is able to quickly adapt a generic pretrained GPT2 to the text generation task at hand.

It is also noted that minimizing the cross-entropy loss (4) is equivalent to minimizing $\mathrm{KL}(\widehat{\boldsymbol{y}}_i^{(SA)} \| \boldsymbol{p}_i^{(GPT2)})$, i.e., the KL-divergence between $\widehat{\boldsymbol{y}}_i^{(SA)}$ and $\boldsymbol{p}_i^{(GPT2)}$, if viewed as distributions over the vocabulary. Due to the asymmetry nature, minimizing the KL-term makes the second slot $\boldsymbol{p}_i^{(GPT2)}$ more wide-spreading than the first slot $\widehat{\boldsymbol{y}}_i^{(SA)}$, illustrated in Figure 1(b). This provides an explanation of why the CE-trained GPT2 could smooth out the noise of the stochastic SA search. As will be shown in experiments, training a GPT2 from SA's output alone can outperform SA.

## 2.3 Sequence-Level Maximum-Margin (MM) Learning

Our next insight is to perform alternations between search and learning to bootstrap performance of TGLS. In the first stage, SA starts local search with the initial candidate being the input (i.e., $y^{(0)} = x$), because we have no other meaningful candidate output yet. Starting with x takes advantage of the resemblance between input and output. But if a higher-quality candidate is available, SA may perform better than from x.

Therefore, we propose another stage of search and learning alternations. SA starts from an initial candidate being GPT2's output, i.e., $y^{(0)} = y^{(GPT2)}$, shown in Figure 1(c). Then, GPT2 is further fine-tuned to learn from the newly searched result. For the learning method, we propose to employ sequence-level max-margin (MM) training, instead of CE training, in this stage. Such alternation can be performed for multiple epochs for performance bootstrapping.

**Algorithm 1:** Training TGLS

**Input:** A non-parallel corpus $X$
**Output:** A fine-tuned GPT2 model
▷ First-stage learning from search
**for** *an input* $\mathrm{x} \in X$ **do**
&emsp;$\mathrm{y}^{(\mathrm{SA})} = \mathrm{SA}(\mathrm{x}, \mathrm{x})$
&emsp;&emsp;&emsp;▷ SA is detailed in Algorithm 2. In the first stage, SA starts with input x as the initial candidate
**for** *all epochs* **do**
&emsp;**for** *an input* $\mathrm{x}$ *with its SA output* $\mathrm{y}^{(\mathrm{SA})}$ **do**
&emsp;&emsp;Fine-tune GPT2 by cross-entropy loss (4) with pseudo-reference $\mathrm{y}^{(\mathrm{SA})}$, conditioned on x

▷ Second-stage learning from search
**for** *all epochs* **do**
&emsp;**for** *an input* $\mathrm{x}$ **do**
&emsp;&emsp;$Y^{(\mathrm{GPT2})} = \mathrm{BeamSearch}(\mathrm{GPT2}(\mathrm{x}))$&emsp;▷ $Y^{(\mathrm{GPT2})}$ is a set of output by beam search
&emsp;&emsp;$\mathrm{y}^{(\mathrm{SA\text{-}S2})} = \mathrm{SA}(\mathrm{x}, \mathrm{y}^{(\mathrm{GPT2})})$ for some $\mathrm{y}^{(\mathrm{GPT2})} \in Y^{(\mathrm{GPT2})}$
&emsp;&emsp;&emsp;▷ In the second stage, SA starts with GPT2's output (any output in the beam is fine)
&emsp;&emsp;$\widetilde{Y} = Y^{(\mathrm{GPT2})} \cup \{\mathrm{y}^{(\mathrm{SA\text{-}S2})}\}$
&emsp;&emsp;Fine-tune GPT2 with max-margin loss (5) with
&emsp;&emsp;&emsp;positive sample: $\mathrm{y}^{+} = \mathrm{argmax}_{\mathrm{y} \in \widetilde{Y}}\, s(\mathrm{y}|\mathrm{x})$, and
&emsp;&emsp;&emsp;negative samples: $\widetilde{Y} \backslash \{\mathrm{y}^{+}\}$

**Return:** Resulting GPT2 (denoted by GPT2-S2 after two stages of search and learning)

Concretely, the GPT2 trained with CE learning performs beam search (beam size $B$) and obtain a set of output sequences $Y^{(\mathrm{GPT2})} = \{\mathrm{y}^{(\mathrm{GPT2},1)}, \cdots, \mathrm{y}^{(\mathrm{GPT2},B)}\}$. A randomly picked (for efficiency purpose) output in $Y^{(\mathrm{GPT2})}$ is taken as initial candidate in SA search, yielding a new sample $\mathrm{y}^{(\mathrm{SA\text{-}S2})}$. We consider the set $\widetilde{Y} = Y^{(\mathrm{GPT2})} \cup \{\mathrm{y}^{(\mathrm{SA\text{-}S2})}\}$ as the positive and negative samples for MM learning. In fact, the positive sample $\mathrm{y}^{+}$ is the best sequence scored by (1), i.e., $\mathrm{y}^{+} = \mathrm{argmax}_{\mathrm{y} \in \widetilde{Y}}\, s(\mathrm{y}|\mathrm{x})$. In most cases, we have $\mathrm{y}^{+} = \mathrm{y}^{(\mathrm{SA\text{-}S2})}$, but this is not necessarily true because SA is not greedy. All other sentences in $\widetilde{Y}$ are collected as negative samples. We use the average of GPT2's pre-softmax logit as the negative energy.[2] In other words, we have $-E(\mathrm{y}) = \frac{1}{N}\sum_{i=1}^{N} z_{i,y_i}$ of a sequence $\mathrm{y} = (y_1, \cdots, y_N)$, where $z_{i,y_i}$ is the logit for the word $y_i$ at the $i$th step. The max-margin loss for this data sample is

$$J_{\mathrm{MM}} = \sum_{\mathrm{y}^- \in \widetilde{Y},\, \mathrm{y}^- \neq \mathrm{y}^+} \max\left\{0, E(\mathrm{y}^+) - E(\mathrm{y}^-) + \Delta\right\}, \tag{5}$$

where $\Delta$ (set to 1) is the margin hyperparameter.

In fact, the energy implicitly defines a globally normalized distribution as $p(\mathrm{y}) = \frac{1}{Z}\exp\{-E(\mathrm{y})\}$, where $Z$ is the partition function. The MM training increases the probability of the positive sample, while decreasing the probability of negative ones. In our MM training, the negative samples are given by beam search on GPT2, highly resembling the positive one. This makes TGLS more sensitive to the sequence-level scorer (1) in its probable region of the output space, illustrated in Figure 1(d).

By contrast, word-level CE increases the probability of the target (analogous to the positive sample) step-by-step, while decreasing the probability of other samples due to local normalization. Thus, it cannot explicitly correct the prediction of a highly-probable but low-scored sample, and performs worse than MM in the second stage.

In summary, the training of TGLS involves two stages of search and learning, where CE and MM are used as the learning objective in different stages. Notice that, for the second stage, search and learning are alternated within the epoch loop. Thus, another stage of search and learning is unnecessary, because our second stage already allows multiple epochs for performance bootstrapping. For inference, we do not perform SA search, but directly use the fine-tuned GPT2 for autoregressive prediction. Appendix B further provides a detailed diagram of our TGLS.

### 2.4 Discussion: TGLS vs. Reinforcement Learning and Structured Prediction

One of the most popular algorithms of reinforcement learning (RL) in text generation is the RE-INFORCE, which maximizes the expected reward (such as the BLEU score [34] or an adversarial discriminator [53]) by sampling a sequence of actions from its learned policy and reweighing the likelihood training of the sampled sequence. REINFORCE is known to have high variance, and previous REINFORCE-based text generation involves groundtruth pretraining [53]. Without a warm start, the sampling-based REINFORCE does not work with such a large action space as the vocabulary. Our TGLS would also optimize an external scoring function (analogous to the reward in RL), but does not have grountruth for pretraining. We instead perform SA search and learn from SA's (local) optima step-by-step.

Monte-Carlo Tree Search (MCTS) [39] is another paradigm of search and learning, where a search tree is maintained with each non-leaf node being a partial configuration (e.g., a partial sentence in text generation). Again, it suffers from the large branching factor, which is the vocabulary size in our applications. Our TGLS adopts local search, which maintains a full configuration and evaluates the candidate at each search step. The resemblance between input and output also largely eases the search task.

The Learning-to-Search (L2S) framework has been successfully applied to various NLP applications, such as structured prediction [6, 4] and text generation [48, 54]. L2S allows the model to explore/search in the space, collects the score (cost) for possible actions, and optimizes the model. Usually, L2S assumes that an expert demonstration (groundtruth sequence and/or dynamic oracle) is available as a reference policy. For instance, a LaSO-like algorithm forces the model to search towards the groundtruth sequence; when the groundtruth is out of the search range, a learning update is performed, where the search effort serves as the negative samples and the groundtruth as positive examples for learning [6, 48]. By contrast, TGLS does not have groundtruth, but uses a strong search algorithm to find higher-scored sentences, which serve as positive samples.

Our approach is also related to learning an inference network for energy-based structured prediction [42, 43]. They perform adversarial learning on the energy model (analogous to a discriminator) and the inference network (analogous to a generator), with the access of groundtruth target. We instead face an unsupervised setting, where we define the heuristic scorer for discrete search; our conditional generator further learns from the search results.

## 3 Experiments

### 3.1 Datasets and Settings

**Paraphrase Generation.** Paraphrase generation is to rephrase input text with different expressions, while keeping the semantics. Following previous work [8, 12], we conducted experiments on the Quora benchmark dataset.[3] We followed the unsupervised setting in [23] and used 500K sentences to fine-tune GPT2 and RoBERTa for fluency and semantic scorers. For validation and testing, we had 500 and 170K samples, respectively.

We adopt BLEU and iBLEU as evaluation metrics, which are widely used for paraphrase generation. BLEU measures the length-penalized $n$-gram overlap between an output and the reference. In addition, paraphrasing requires that the output should be different from input. Thus, iBLEU [40] penalizes BLEU by $n$-gram similarity between output and input. Following most work, we consider iBLEU as the main metric for paraphrasing.

**Text Formalization.** This task concerns formality transfer of text, and our goal is to rephrase a given informal text into the formal style. We experimented with the Grammarly's Yahoo Answers Formality Corpus (GYAFC) [35] in the domain of Family & Relationships. It is noted that GYAFC contains 50K informal–formal pairs, but our TGLS follows the setting of most other style-transfer work [14], which uses non-parallel corpora with style labels, but does not have parallel supervision. Our pretrained language models are additionally fine-tuned on automatically labeled non-parallel corpus [50]. In GYAFC, there are 3K samples for validation and 1K for test.

Table 1: Automatic evaluation results on paraphrasing.

| Methods | iBLEU | BLEU |
|---|---|---|
| Supervised | | |
| RL-NN [32] | 14.83 | 20.98 |
| DAGGER[†] [7] | 18.88 | 28.42 |
| GPT2[†] [33] | 19.19 | 26.92 |
| Distant supervised | | |
| Round-Trip MT (GPT2)[†] [11] | 11.24 | 16.33 |
| Round-Trip MT (Transformer)[†] [26] | 14.36 | 20.85 |
| Unsupervised | | |
| VAE [3] | 8.16 | 13.96 |
| CGMH [28] | 9.94 | 15.73 |
| UPSA [23] | 12.02 | 18.18 |
| SA w/ PLM (Ours)[†] | 14.52 | 21.08 |
| TGLS (Ours)[†] | **17.48** | **25.00** |

Table 2: Automatic evaluation results on formality transfer. $\downarrow$The smaller, the better.

| Methods[†] | PPL$\downarrow$ | BLEU | Formality | H-mean | G-mean |
|---|---|---|---|---|---|
| Supervised | | | | | |
| LSTM-attn [35] | **23.42** | **69.36** | **87.39** | **77.34** | **77.85** |
| Unsupervised | | | | | |
| BackTrans [31] | 183.7 | 1.23 | 31.18 | 2.37 | 6.13 |
| StyleEmb [9] | 114.6 | 8.14 | 12.31 | 9.80 | 10.01 |
| MultiDec [9] | 187.2 | 13.29 | 8.18 | 10.13 | 10.42 |
| CrossAlign [38] | 44.78 | 3.34 | 67.34 | 6.36 | 14.99 |
| DelRetrGen [21] | 88.52 | 24.95 | 56.96 | 34.70 | 37.69 |
| Template [21] | 197.5 | 43.45 | 37.09 | 40.02 | 40.14 |
| UnsupMT [56] | 55.16 | 39.28 | 66.29 | 49.33 | 51.02 |
| DualRL [25] | 66.96 | 54.18 | 58.26 | 56.15 | 56.18 |
| TGLS (Ours) | **30.26** | **60.25** | **75.15** | **66.88** | **67.29** |

† indicates that the results are directly comparable to TGLS on the same data split. Appendix C provides more details on the baseline models and how these results are obtained.

The performance of formality transfer is measured in different aspects. The language modeling perplexity evaluates the fluency of the generated text, and a separately trained classifier predicts the formality accuracy. Particularly, the formality evaluator achieves an accuracy of 94%, being a good automatic evaluation measure.[4] The BLEU score is also computed against the reference to evaluate $n$-gram overlap. Finally, we consider the harmonic mean (H-mean) and the geometric mean (G-mean) of the formality accuracy and the BLEU score as our main metrics for this task.

**Hyperparameters.** For SA, the initial temperature was set to 1e-2 in both tasks. The total search steps and temperature cooling were 50, 2e-4 for paraphrasing; and 100 and 1e-4 for text simplification. The scorers' weights were tuned by grid search, set as $(\alpha, \beta, \gamma, \delta) = (0.8, 1, 0.6, 0.125)$ for paraphrasing, and $(0.8, 2, 1.25, 0.26)$ for text formalization. We keep the RoBERTa fixed and further tune the GPT2 model by alternations of search and learning for another 6 epochs.

## 3.2 Overall Performance

Table 1 presents the results of automatic evaluation for paraphrase generation. Among the unsupervised approaches, the simulated annealing model UPSA [23] achieves the previous state-of-the-art performance, outperforming both variational sampling [3] and discrete-space Metropolis–Hastings sampling [28]. We propose to use large-scale pretrained language models for fluency and evaluation (denoted by SA w/ PLM), and improve iBLEU by 2.5 points from UPSA. Our TGLS framework of search and learning further improves iBLEU by 2.96 points, being a new state-of-the-art unsupervised paraphrasing model.

The TGLS also outperforms the paraphrasing systems based on round-trip translation, which is widely used in real-world applications. Such methods generate a paraphrase by translating a sentence to a foreign language and translating it back. It is categorized as distant supervision, because it requires parallel corpora for machine translation, but not for the paraphrasing task of interest.

Noticeably, our unsupervised TGLS performs comparably to a few recent paraphrasing model [32, 7]. Moreover, we train a GPT2 in the supervised setting for a controlled experiment, where the neural architecture is fixed. We see that the unsupervised TGLS is slightly worse than the supervised setting by only 1.71 iBLEU, largely closing the gap between supervised and unsupervised paraphrasing.

Table 2 presents the results for formality transfer. Again, we see consistent evidence on the effectiveness of TGLS, as it outperforms existing unsupervised approaches including heuristic marking of style words and retrieval-based editing [21], unsupervised machine translation approaches [56], and dual reinforcement learning [25].

Admittedly, the unsupervised TGLS is still worse than supervised approaches on this task. This is probably because our heuristic scorers are mostly designed for the paraphrasing task, and even for

Table 3: Model analysis on paraphrase generation. All variants use pretrained language models.

| Methods | iBLEU | BLEU | Inference Time (sec/sample) |
|---|---|---|---|
| SA | 14.52 | 21.08 | 5.46 |
| SA+CE | 14.97 | 23.25 | 0.06 |
| SA+CE+SA | 15.41 | 21.48 | 2.62 |
| SA+CE+SA+CE | 15.70 | 21.70 | 0.37 |
| SA+CE+SA+MM (full) | **17.48** | **25.00** | 0.43 |

large-scale pretrained models, their performance may drop with informal text. More effort could be made here for future work.

We also conducted human evaluation, reported in Appendix D. Results are consistent with these automatic metrics.

## 3.3 Analysis

In this part, we present an in-depth analysis of our model with paraphrase generation as the testbed.

**Ablation study.** As TGLS involves two stages of search and learning, we conduct an ablation study, shown in Table 3. We start from a base simulated annealing (SA) approach, where we have already adopted pretrained language models. Thus, it sets up a fair comparison.

In the first stage of learning, our GPT2 model with word-level cross-entropy (CE) training already outperforms SA alone. The result is slightly surprising, but it actually makes sense because cross-entropy loss can smooth out the noise in SA's heuristically defined search objective.

We also tried to train the GPT2 by max-margin (MM) loss without CE learning, but it fails to escape from a random policy. It is due to the difficulty of training an energy-based model in comparison to a locally normalized model [10]. In our work, the negative samples in the beam would be useless when the model is not warm started.

We compare SA with the initial sentence being input and GPT2's prediction (SA vs. SA+CE+SA). We see the latter outperforms both SA and SA+CE. This confirms that the learned GPT2 helps SA find a better optimum.

The last two lines of Table 3 provide evidence of performance bootstrap by alternating between search and learning, as they outperform other ablated variants. In particular, MM is better than CE by a significant margin in the second stage. Our intuition is that MM with negative samples in the beam makes TGLS more sensitive in distinguishing sentence quality with its highly probable output region.

**Efficiency analysis.** We report inference time in Table 3. The experiments were conducted on a cluster with Nvidia Telsa V100 GPUs. The inference time could be noisy due to the multi-thread nature of clusters, but it provides a conclusive enough comparison between search-based and autoregressive generation. As seen, SA is inefficient because it requires hundreds of steps of editing and reevaluation. SA+CE, SA+CE+SA, SA+CE+SA+CE, and SA+CE+SA+MM are all based on the GPT2 model during inference, and thus are much more computationally efficient. Based on the validation, SA+CE adopts greedy decoding, whereas the others adopt beam search with a size of 5. We see all GPT2-based generators are at least $6\text{--}10\times$ faster than the search-based methods.

The training efficiency of TGLS is roughly twice as much as SA plus GPT2 fine-tuning. We do not have quantitative comparison, because training efficiency highly depends on hyperparameters and early stop strategies. While our training is more complex than SA or GPT2, we do not view it as a disadvantage. First, training is usually done offline; when trained, our model is very efficient for deployment compared with SA. Second, it is understandable that we sacrifice some training efficiency compared with supervised models, since we do not have parallel data. In fact, our approach should be more efficient (and labor-saving) than data collection plus human annotation in the supervised setting, as explained in "Broader Impact."

**Case study.** We present a case study in Appendix E. Typical examples show that TGLS is able to generate more fluent and more different-appearing paraphrases than search-based methods.

# 4 Related Work

**Unsupervised text generation**. One popular approach to unsupervised text generation is the variational autoencoder [15], which generates text by manipulated latent space for certain attributes, such as sentiment [14], topic [45], and syntax [55].

Recently, search-based methods have been developed for various text generation tasks, including text simplification [17], summarization [37], keyword-to-text generation [28], and paraphrasing [28, 23]. However, these methods are not learnable; hence, the inference is inefficient and the performance cannot be improved by training.

Most of other work of unsupervised text generation is built upon heuristics of a certain task. For instance, Narayan and Gardent [29] propose a task-specific pipeline for sentence simplification. Zheng and Lapata [58] employ a graph-based ranking algorithm to select the most significant sentence as the summarization of a document. Chu and Liu [5] utilize the overlapping of text as a hint for multi-document summarization. In our work, the proposed TGLS is a principled search-and-learning framework, where it is also possible to encoding prior knowledge of a task into the search algorithm.

**Paraphrase generation.** Recent progress on paraphrase generation is largely due to neural models trained with large-scale parallel data. Researchers have applied search-and-learning approaches for supervised paraphrasing, such as reinforcement learning (RL) [22, 32, 52] and learning-to-search (L2S) [7] (see Section 2.4 for discussion on these machine learning models). Our approach is in the unsupervised setting.

Researchers have proposed roundtrip translation for paraphrasing, i.e., translating a source sentence into a pivot language, and then translating it back into the original language [57, 26, 11]. Although no supervision of paraphrases is needed, the success of this approach depends on high-quality machine translation (MT) systems, hence requiring large-scale parallel MT datasets. This can be thought of as distant supervision for paraphrasing.

In the unsupervised setting, paraphrases can be generated by either a variational latent-space sampler [2] or a word-space Metropolis–Hastings (MH) sampler [28]. By decreasing the temperature of the stationary distribution, Liu et al. [23] show that search-based formulation outperforms sampling for unsupervised text generation. Our work further extends it to the learning-from-search framework, improving both accuracy and inference efficiency.

**Text style transfer.** Our text formalization is one application of text style transfer. Other examples include sentiment [14] and the prose style [51]. Typically, text style transfer can be divided into three categories: parallel supervised, non-parallel supervised (with only style labels), and purely unsupervised. Parallel supervised style transfer trains a sequence-to-sequence model [35], whereas purely unsupervised style transfer replies on disentangling latent space [49].

Most previous work on text style transfer is in the non-parallel supervised setting, assuming style labels are available. Researchers have developed style embedding-based approaches [38, 9], style-specific decoders [9], style-word editing approaches [21], among others. Our approach also follows this setting, but to the best of our knowledge, we are the first to model style transfer as a search problem, as well as to extend it to the proposed TGLS framework of learning from search.

# 5 Conclusion

This work proposes a novel learning-from-search framework TGLS to unsupervised text generation. We show that the simulated annealing search can provide high-quality examples for training a conditional text generator. Further, the generative model can give a better initial state to the search algorithm. Experiments demonstrate that the alternation of search and learning can boost the performance of TGLS on two unsupervised text generation tasks, paraphrase generation and text formalization. Moreover, our model is considerably more computationally efficient, compared with search-based generation methods. We note that TGLS opens a few future directions, such as more effective and efficient search algorithms, more noise-robust learning methods, and a better combination of search and learning. We would also like to apply the learning-from-search framework to other sequential prediction tasks in NLP.

# 6 Broader Impact

Our TGLS is a new framework of search and learning for natural language processing (NLP) applications. Typically, an NLP application is different from an agent-based reinforcement learning (RL) setting, as our "environment" is fixed. Previous work in NLP usually uses the REINFORCE algorithm, which introduces much noise and is inefficient because actions are sampled from its policy. Imitation learning (IL), another framework for search and learning, is successful in various NLP structured prediction tasks, but requires expert demonstrations (i.e., groundtruth). Our work alleviates the drawbacks of RL and IL by injecting a strong search algorithm in the search-and-learning loop. With the success in paraphrasing and text formalization in our paper, we expect TGLS would be potentially applicable to other NLP tasks.

For social aspects, the unsupervised nature of TGLS would have the following potential impact.

- **Reducing human annotation labors.** Most text generation approaches need parallel data for supervised training, which oftentimes in turn requires human annotation. Our TGLS does not rely on annotated parallel data. Experiments show that TGLS largely reduces the performance gap between supervised and unsupervised text generation, sometimes even comparable to recent supervised models. This reduces the need of human annotation in the research and application of text generation.

- **Helping low-resource language processing.** Previous text generation studies hardly tackle low-resource language, due to the lack of parallel corpora. Our unsupervised TGLS is easily applicable to low-resource language, thus benefiting people from different cultures speaking different languages.

- **Supporting small businesses and new applications.** Small businesses, different from a large company, may not have enough human and financial resources to annotate large-scale corpora. They may apply our unsupervised TGLS as a starting point for a new project, or even a new application.

**Acknowledgments.** The work described in this paper was supported by the National Key Research and Development Program of China (No. 2018AAA0100204), the Research Grants Council of the Hong Kong Special Administrative Region, China (No. CUHK 14210717 of the General Research Fund), the Alberta Machine Intelligence Institute (Amii) Fellow Program, the Canadian CIFAR AI Chair Program, and the Natural Sciences and Engineering Research Council of Canada (NSERC) under Grant No. RGPIN-2020-04465. This research was also enabled in part by the support of Compute Canada (www.computecanada.ca).

## Footnotes

[1]Code is available at https://github.com/jingjingli01/TGLS

[2]*Energy* is what MM learning would like to minimize for positive samples.

[3]https://www.kaggle.com/c/quora-question-pairs

[4]We reuse the architecture of RoBERTa for formality evaluation and GPT2 for fluency evaluation. However, they are separately trained, third-party models, and are NOT part of our TGLS.

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
