[Supplementary Material]

# A Fine-Tuning Language Models for Scorers and Edit Proposal

Firstly, we describe the strategy of fine-tuning RoBERTa, which is used for scoring semantic compliance and the proposal of replacing words in simulated annealing (Section 2.1). In particular, we consider two fine-tuning objectives, as follows.

**Masked language model.** This fine-tuning objective is based on domain-specific unlabeled corpora, and its goal is to adapt RoBERTa and make it more specific to the domain at hand. For each experiment, we use its unlabeled training corpus for fine-tuning.

Generally, we follow the mixed masking strategy [24], which randomly masks out a few words in a sentence, and the fine-tuning goal is to predict these masked words. The mixed masking strategy randomly picks one of the three types of masking: (1) with probability 80%, the input is substituted by a special token [MASK]; (2) with probability 10%, the input is substituted by a random word in the vocabulary; and (3) with probability 10%, the input is substituted by the word itself (i.e., no masking is performed).

We observe that the first mask type aligns with the `Replace` and `Insert` proposals of local editing. Thus, we would high weigh this mask type in our fine-tuning. Each time we process a data sample, we perform one more masking with the special token [MASK], in addition to the mixed strategy.

**Formality classification.** This objective is specific to the text formalization experiment, where RoBERTa is also used for training a formality classifier. The objective is cross-entropy loss between $p(\text{formal} \mid \text{RoBERTa}(\text{BOS}, \text{x}))$ and the groundtruth formality label (formal or informal), where $\text{RoBERTa}(\text{BOS}, \text{x})$ is the RoBERTa feature of a sentence $\text{x}$ in the unlabeled dataset. Still, no parallel corpus is used. This fine-tuning objective works together with the masked language modeling objective in a multi-task fashion.

Note that the formality classification objective does not apply to the paraphrasing task.

**Conditional generator.** We fine-tune the GPT2 model with task-specific data. For paraphrasing, we use all training sentences, whereas for style transfer, we use the training set of the formal style only.

Figure 2: Two stages of search and learning in TGLS.

**Algorithm 2:** SA for Text Generation [23]

**Input:** An input sentence x,
 An initial candidate output $y^{(0)}$
**Output:** An output sentence y
**for** $t = 1, \cdots, \text{MaxStep}$ **do**
 Set temperature $T = \max\{T_{\text{init}} - C \cdot t, 0\}$    $\triangleright T_{\text{init}}, C$: annealing hyperparameters
 Randomly pick an edit operator $\text{Op} \in \{ \texttt{Delete, Insert, Replace}\}$
 Randomly pick an edit position $k$
 Propose a new candidate $y' = \text{Op}(y^{(t-1)}, k)$
 Compute acceptance rate $p(\text{accept}|y', y^{(t-1)}, x, T) = \min\{1, \exp(\frac{s(y'|x)-s(y^{(t-1)}|x)}{T})\}$
$$y^{(t)} = \begin{cases} y', & \text{with probability } p(\text{accept}|y', y^{(t-1)}, x, T) \\ y^{(t-1)}, & \text{with probability } 1 - p(\text{accept}|y', y^{(t-1)}, x, T) \end{cases}$$
**Return:** $y^{(\text{SA})} = \text{argmax}_t\, s(y^{(t)}|x)$

For fine-tuning hyperparameters, we performed 3 epochs of fine-tuning for text formalization and 9 epochs for paraphrasing. The maximum length of input was set to 35. We use Adam with $\beta_1 = 0.9$ and $\beta_2 = 0.999$ for optimization.

## B Diagram of TGLS

Figure 2 shows the diagram of TGLS. Algorithm 1 further presents the pseudo-code of SA search [23] for reference.

## C Competing Models

We present more details on the competing methods in Tables 1 and 2. All metrics are computed based on word-level tokenization (i.e., no BPE segmentation is used).

**Paraphrase Generation**

- **RL-NN.** Qian et al. [32] propose to learn a reward function by neural networks and perform REINFORCE training. The results in Table 1 are from the original paper.

- **DAGGER.**[†] Du et al. [7] apply imitation learning to paraphrase generation. It achieves the state-of-the-art performance on the Quora dataset. We re-ran their implementation based on our data split.

- **GPT2.**[†] We train another GPT2 with the same hyperparameters as our TGLS, but in a supervised setting for a controlled comparison.

- **Round-Trip MT (Transformer).** Following [26], we utilize a well-trained bi-directional neural machine translation (NMT) system (Zh→En and En→Zh) with a Transformer model [44]. The NMT system achieves BLEU scores of 43.2 (En→Zh) and 28.74 (Zh→En) on the Newstest 2017 dataset. In our work, we use the round-trip translated sentence (En→Zh→En) as the paraphrase.

- **Round-Trip MT (GPT2).** Similarly, we adopt another GPT2-based multilingual (En, Zh, Es, Ru) NMT system in [11]. Suggested by [11], we take the Zh as the pivot language.

- **VAE.** The variational autoencoder [3] generates a paraphrase by sampling from the encoded posterior distribution in the latent space. Here, we quote the results of CGMH from the implementation of [23].

- **CGMH.** Miao et al. [28] propose a word-space Metropolis–Hastings approach to paraphrase generation. Results are also quoted from the implementation of [23].

- **UPSA.** Liu et al. [23] extend CGMH by decreasing the temperature and this becomes simulated annealing. Results are quoted from the original paper.

Table 4: Human evaluation on the Quora dataset.

| Method | Grammar, Fluency | Coherency, Consistency | Agreement |
|---|---|---|---|
| UPSA [23] | 4.05 | 3.28 | 35.0% |
| SA w/ PLM | 4.79 | 4.48 | 70.0% |
| Our TGLS | **4.85** | **4.66** | 78.8% |

- **SA w/ PLM.**[†] One of our extensions to UPSA is to fine-tune pretrained language models for the search objective and edit proposals. This variant is essentially the intermediate results of our TGLS, after its first-stage SA search.

While widely used for paraphrasing, the Quora dataset does not contain a standard split. The dataset is crawled from the Internet, and thus it is noisy and sometimes contains duplicate samples in training and test sets. This would not be a severe problem if the duplication is between training input and reference output during supervised learning; thus, most previous work does not explicitly deduplicate these samples. However, this could affect our TGLS, because we perform learning from search results with the non-parallel training set. Thus, we carefully handled this problem, ensuring no overlap in training and test.

The competing models with † indicate that the data split is the same as TGLS, and the results are directly comparable. Others can be compared in a statistical sense.

**Text Formalization**

- **LSTM-attn.** Rao et al. [35] trained a Bi-LSTM encoder-decoder model with attention on their parallel formality corpus.
- **BackTrans.** Prabhumoye et al [31] utilizes back-translation to get style-independent content representations and feed them to style-dependent decoder to control the style of output.
- **StyleEmb.** Fu et al. [9] propose two variants for style transfer. In this variant, they accomplish style transfer by a learned style embedding.
- **MultiDec.** The other variant of [9] use multiple decoders for style-specific generation.
- **CrossAlign.** Shen et al. [38] also use style embedding, but they apply adversarial training based on style-transferred hidden states to cross-align content.
- **DelRetrGen.** Li et al. [21] propose a heuristic approach to mark style-specific words and phrases, and obtain expressions in a desired style by retrieval. Eventually, a neural model generates a style-transferred sentence.
- **Template.** This is a simpler variant in [21]. Then the detected style-specific words of input sentences are replaced by stylized words of target domain within its retrieved counterpart.
- **UnsupMT.** Zhang et al. [56] apply unsupervised machine translation techniques for style transfer. They firstly conduct word-to-word transfer and construct pseudo sentence pairs for system initialization, then conduct iterative back-translation training.
- **DualRL.** Luo et al. [25] use a dual reinforcement learning strategy to learn bi-directional style transfer without explicitly separating the style and content.

The results in Table 2 involve learnable metrics. We used separately trained GPT2 and RoBERT for fluency and formality evaluation, respectively. The GYAFC corpus has a standard dataset split. For fairness, we re-evaluated all the outputs based on our own evaluation models.

The outputs of LSTM-attn are released by [35], and the rest outputs are published by [25].

## D  Human Evaluation

We conduct human evaluation for the paraphrase generation experiment with selected baselines that are most relevant to our work, due to the limit of budgets. We randomly selected 120 paraphrase samples. Four annotators with linguistic background were invited to assess each sentence pair on a scale of 1–5 with regard to two criteria: relevance (coherence & consistent) and naturalness (grammar

Table 5: Examples generated by SA (w/ PLM) and the full TGLS model.

| # | Source | SA w/ PLM | TGLS |
|---|--------|-----------|------|
| 1 | how better is sony mdr-xb950ap than sony mdr-xb450? | how better is the new sony mdr-xb or the sony mdr-xb? | how good is the sony mdr-xb950 vs the sony mdr-xb450? |
| 2 | what is meant by british political conservatism? | what is meant by the british? | what is the british political conservatism? |
| 3 | what are the problems occur when flow velocity increases in a pipe? | what are the problems that arise when the velocity of an increase in a cylinder increases? | what are some problems when the flow velocity of a pipe increases? |
| 4 | if a person gets out of a vehicle can they still receive a citation? | can a person gets off of a vehicle, do they still receive a citation? | can a person get out of a car without a citation? |

& fluency) in a blind manner. Each example was reviewed by two judges and average scores are reported in Table 4. We also show the percentage of agreement between human annotators. As seen, humans have a higher agreement when the model performance is high; otherwise, the agreement is lower, because different annotators have their own scale among 1–5.

In terms of performance, our extension of UPSA—with pretrained language models for fluency and semantic scorers—indeed improves human satisfaction in terms of Grammar/Fluency and Coherence/Consistency. Our TGLS model further improves the performance in both aspects. The results are also consistent with the automatic metrics in Section 3.2.

# E   Case Study

We show in Table 5 examples generated by SA (with pretrained language models) and the full TGLS. As seen, SA sometimes does not generate fluent sentences. In Example 2, the phrase "*political conservative*" is deleted but no synonyms are suggested as a replacement. Our TGLS is able to generate more fluent sentences. Moreover, our TGLS generates a paraphrase in an autoregressive fashion, thus sometimes yielding a more different-appearing output, e.g., "*flow velocity increases in a pipe*" being rephrased to "*flow velocity of a pipe increases*" in Example 3.

In Example 4, we also see that TGLS generates a seemingly plausible paraphrase given the input. However, the output conveys an opposite intention to the input. This shows that understanding the logic and pragmatics of language is still difficult for large-scale pretrained language models, and deeper semantic analysis shall be addressed in future work.