[Reviews · NeurIPS 2020]

Review 1

Summary and Contributions: This paper proposes a new method for unsupervised text generation by search and learning, where it explores the output space to find good candidates (measured by a score function) to form a pseudo-parallel dataset, then a GPT2 model is fine-tuned on it with cross-entropy loss to be able to make predictions. The outputs of the fine-tuned GPT2 model are used as the starting point to explore the output space again to form a better pseudo-parallel dataset. Then the GPT2 model is fine-tuned again but with sequence-level maximum-margin training which is shown to obtain much better results. Experimental results on unsupervised paraphrasing and text formality transfer are strong, and the inference is much simpler and faster than previous search-based baselines. ---------- After response ---------- Thank you for the response! I still feel like the authors should be cautious to claim it as generic text generation framework -- so far this line of work is more like unsupervised style transfer including the mentioned related work of grammar correction or sentence summarization. These are tasks that many style transfer papers actually evaluate on.

Strengths: (1) The method is novel to tackle unsupervised text generation. Despite the somewhat complicated training pipeline, the inference is as simple as decoding from a GPT2 model which is much faster than the previous SA baseline (2) The empirical results on unsupervised paraphrasing and text formality are strong

Weaknesses: (1) This paper claims the method as a "generic unsupervised text generation framework", but only evaluates on relatively easy tasks like paraphrase generation and formality style transfer where many words are shared between the input and output, and there are other very standard and representative sequence generation tasks like machine translation or summarization. The related work section also only discusses literature on these two tasks. Thus I feel like the paper (especially the intro) is a bit over-claiming -- the paper needs to either include other standard tasks to make the current claim valid, or downplay the claim and discuss the limitations of the proposed method in a separate (sub)section. For example, I would expect it is much more difficult for SA to explore the output space for machine translation, summarization, dialogue, etc., which means the proposed method may not be "a generic framework". I think paraphrasing and formality transfer are very special cases where SA can work well, thus I can see many limitations of the proposed method unless the authors demonstrate enough experimental evidence. (2) In Table 3 it seems MM accounts for a large part of the gain, I wonder whether “SA + MM” would suffice or not.

Correctness: Yes

Clarity: Yes

Relation to Prior Work: Yes

Reproducibility: Yes

Additional Feedback: (1) The subcaptions in Figure 1 are duplicated ? (2) The paper uses the language model GPT2 to conduct a sequence to sequence task, it is interesting to see how a sequence-to-sequence model would work in replace of GPT2.


Review 2

Summary and Contributions: The paper presents an unsupervised text generation technique by search and learning. Authors show effectiveness of their proposed approach on two NLG tasks: paraphrase generation and text formalization. Even though search based unsupervised text generation technqiues already exist but not without limitation. This paper aims to address limitation like noisy search result, low inference efficiency. They also consider integration of large-scale language model into their technique as their contribution.

Strengths: Extensive experimental evaluation of their unsupervised text generation technique. Better performance than current SOTA unsupervised techniques for paraphrase generation and text formalization. Idea of local edits is interesting for unsupervised paraphrasing and text formalization even though it is already applied in grammar correction etc.

Weaknesses: The papr has limited novelty. fluency scorers, task specific scorers, RL based training etc is already covered in previous literature for paraphrase generation. It would have been interesting to see does it generalizes to other complex tasks like question generation etc.

Correctness: Evaluation methodology is correct. It would have been interesting to see human evaluation results, as BLEU might not be the best evaluation metrics.

Clarity: Paper could have been written in a bit better manner. Overall how each bits and pieces of the text generation framework stiches together and trained it not very clear after reading the paper.

Relation to Prior Work: Paper discuss prior work. But it is not very clear how the technique differs from previous unsupervised search based text generation techniques. Component wise comparision of unsupervised text generation techniques is missing.

Reproducibility: No

Additional Feedback: The idea of unspervised text generation is good. why the technique has been evaluated on only two relatively simpler tasks? Why not try this out on nmt, Question generation etc? The overall approach section could be organised in a better manner. It would be interesting to see some human evaluation results. How do you that the discussed fluency and semantic scorer is good enough? is there any way to evaluate them?


Review 3

Summary and Contributions: This paper proposes a novel learning framework TGSL for unsupervised Text Generation. It first use a search-based generator as a teacher to train GPT2. Then, the outputs of GPT2 are taken as the initial state of the search algorithm again for iterative performance improvement, and they perform max-margin (MM) learning to better distinguish between higher-scored sentences and other high-probability sentences. Experiment results show their model can achieve comparable performance on two real-world natural language generation tasks comparing with supervised methods. My intuition is that the motivation of this paper is not clear. It only replaces the original scoring function with a more powerful pre-trained model and the search-based method does not give details.

Strengths: (1) A novel unsupervised text generation framework for text generation. (2) TGSL is effective for paraphrase generation and text formalization. (3) I like the demonstration in Algorithm 1 that clearly depicts the overall process of TGSL.

Weaknesses: (1) As I mentioned in the summary, the motivation of this paper is not strong. I don’t know for what purpose the author designed a very complex unsupervised generation framework. (2) What is the search-based generator algorithm? How to be unsupervised? Unknown to a reviewer. (3) The paper lacks the conclusion section

Correctness: There is no problem with the method of the paper, but the innovation is slightly insufficient

Clarity: Although some details of this paper are not explained clearly, the overall is still very good.

Relation to Prior Work: The reference is summarized very well.

Reproducibility: Yes

Additional Feedback:


Review 4

Summary and Contributions: This paper builds upon previous work [1] on using simulated annealing for paraphrase generation which was completely search based, to also make use of pre-trained language models as part of the scoring process as well as for bootstrapping the generation. They extend the fluency and semantic scorers by incorporating GPT-2 and Roberta to calculate these scores. This paper shows improved results over previous unsupervised baselines for paraphrase generation as well as text formalism. They use the outputs of the SA algorithm as a pseudo-reference for a synthetic language modeling task using GPT-2 where the model is fine tuned to predict the pseudo reference. The prediction from this model is then used as a seed for further steps of search. After the second search, a max margin loss is applied to the generations to distinguish between the best scoring generation under the search and the rest.

Strengths: Gains over previous unsupervised methods for the task of paraphrase generation and text formalization.

Weaknesses: 1) The several stages would introduce a lot of additional compute and latency which is not discussed. 2) Given that this model incorporates GPT-2 and RoBERTa, it would be necessary to compare to baselines with comparable number of parameters. 3) Inference time speedups are claimed with respect to their own search based methods, but would be meaningful if given in context of inference times of other baseline methods.

Correctness: Needs an analysis of training time and tradeoff of the different approaches with respect to the number of iterations of search required.

Clarity: Better fluency in explaining the different parts of the method would be great. It was a bit confusing to read about all the different losses before the description of how they all plug in together. Algorithm 2 which describes the main search step of the paper needs to be in the main paper and not in the appendix Minor typos: 1) 81 - bype-pair -> byte-pair 2) 208 - "as unsupervised text generation"-> "as for unsupervised text.."

Relation to Prior Work: Yes

Reproducibility: Yes

Additional Feedback: Why use greedy decoding for SA+CE vs beam search for the other two?


Review 5

Summary and Contributions: This paper claim that they propose a framework named TGSL for unsupervised text generation by search and learning. They also demonstrate their model outperform previous unsupervised approach on two tasks: paraphrase generation and text formalization.

Strengths: In this paper, the authors use powerful pre-trained language models RoBERTa and GPT2 in their framework for better search and generation. And the proposed framework largely outperforms previous unsupervised approaches. It is great that the authors take a lof efforts to make the framework work well.

Weaknesses: 1. Some detaills are missing as I mention below. I undertand this situation becuase of the lots of details for the experiments. Hope it will be more clear in the next version. 2. I am not sure whether the method are called search-based. It seems like it is simulated annealing for approximating the global optimum based on the scores in Equation (1). It is still a sampling-based method. 3. In this framework, the final outputs are from fine-tuned GPT2, not the SA outputs based on the scorer. How is the performance on SA output? In your work, the fine-tuned GPT2 could be treated as the inference network during the decoding. It looks similar to the dirction of these work [1] [2][3]. 4. I think the reader could see the effect of scores. How is the learned scorers? Maybe an ablatioin study could be done? Or compasion bettwen several outputs? [1]. Lifu Tu, Kevin Gimpel. 2018. Learning Approximate Inference Networks for Structured Prediction [2]. LIfu Tu, Kevin Gimpel. 2019. Benchmarking Approximate Inference Methods for Neural Structured Prediction. [3]. Lifu Tu, Richard Yuanzhe Pang, Sam Wiseman, Kevin Gimpel. 2020. ENGINE: Energy-Based Inference Networks for Non-Autoregressive Machine Translation

Correctness: 1. Equation (5) should be max{0, E(y^+) + \Delta - E(y^-)}. 2. In line 163 and line 164, the energy is summation of GPT's pre-softmax logit. Howver, the logit is unnormlized. Maybe the authors refer to the score after log operation?

Clarity: The writting part of this paper could be improved. Some detail about the experimetns are missing: 1. For margin-based training, how do you set the margin cost term \Delta? 2. And in Algoirhtm 1 and Algorithm 2, there is one important hyperparameter: epoch number. Maybe it is good to show the effect of this hyperparameter? 3. In algorithm2, beam search algorithm is used. What is the beam size? What are the effect of the beam size in your algorithm? Do you constrain the length of the outputs? The length seems to have effect on the scorers. 4. What is the denitiion for N in the paper (line 132 and line 164)? In line 132, N seems the length of the length of y.

Relation to Prior Work: "The alternation between search and margin-based learning": Lifu Tu, Kevin Gimpel. 2018. Learning Approximate Inference Networks for Structured Prediction "energy-based or score-based text serarch without ground truth ": Lifu Tu, Richard Yuanzhe Pang, Sam Wiseman, Kevin Gimpel. 2020. ENGINE: Energy-Based Inference Networks for Non-Autoregressive Machine Translation

Reproducibility: No

Additional Feedback:

[Author Response · NeurIPS 2020]

Thank reviewers for detailed comments. Our main contribution is the novel search-and-learning for UnsupTextGen, achieving remarkable performance (sometimes even better than SupTextGen). Our novelty, clarity, and performance are recognized by different reviewers. Despite some borderline score, we find reviewers' concerns can all be resolved and do not diminish our contributions, addressed below.

**R1:** Thanks for recognizing our novelty and performance. **3.1 (generic UnsupTextGen framework):** By "generic" we mean the model can be applied to different tasks that share the same problem structure. In fact, search-based UnsupTextGen has shown promising results for summarization[36], simplification[27,41], grammar correction[26]. Our framework differs from application-specific UnsupTextGen. For example, the rules/heuristics in [18] only applies to sentiment style transfer. **Future work MT:** Thanks for suggesting the future work. We are currently considering MT by using word-level dictionary and performing search and learning. We're happy to revise the terminology and highlight what applications TGSL is appropriate for, namely, those where the input and output show certain resemblance. We expect, however, this can be relaxed by using different search algorithms other than local search. **3.2 (SA+MM):** We mentioned in Line 269 that SA+MM cannot achieve reasonable performance. This is because MM is hard to train without warm start (training signal is too sparse merely by several negative samples).

**R2: 3 (Novelty):** Our main novelty is the search-and-learning framework TGSL for UnsupTextGen, where our learning is non-trivial and involves two stages with different losses, well motivated and supported by ablation study. To our best knowledge, we are the first to work in this direction. The scorers highlighted by the reviewer are additional engineering contributions of this paper; TGSL also largely differs from all previous RL models (see §2.4 for detailed discussion). **4 (no human eval):** human eval is in Appendix D, summarized in Line 258, main paper. **6 (How...differs from previous unsupervised search TextGen):** We propose a search-and-learning framework[§2.2-2.3] for UnsupTextGen; previous work is search only. **Component wise comparison of UnsupTextGen:** We're unsure what "component wise comparison" is; Tab 3 presents a rigorous ablation study of our model with previous work, where model capacity is strictly controlled. **8 (future work MT/QuestionGen):** See response to R1. Thanks!

**R3: 1 (only replaces original scoring):** Our main contribution is the search-and-learning framework[§2.2-2.3]. Improving the original scorer[§2.1] is our additional contribution. **3.1 (motivation):** UnsupTextGen has wide applications, e.g., low-resource language and cold-start for new applications, where large-scale parallel corpora are unavailable[Line16]. Motivation for search-and-learning (main contribution) is to improve efficiency by avoiding search and to improve performance by smoothing out search noise[Line30]. **3.2 (search-based UnsupTextGen):** Search component is simulated annealing (SA)[Line66]. UnsupTextGen is feasible with a manually defined objective function for the strong SA search. We used SA but further proposed the novel search-and-learning framework. **3.3 Conclusion (and clarity):** Thanks for saying the overall clarity is "very good." Due to space limit, we omitted Conclusion for submission. We'll expand the details and include the conclusion should the paper be accepted.

**R4: 3.1 (additional compute/latency):** Our inference efficiency is as low as a conditional text generator, but 5–10x faster than previous search UnsupTextGen[Tab 3]; training efficiency $\approx 2\times$(SA+finetuneGPT2). **3.2 (controlling #parameters):** Yes, the number of parameters is strictly controlled in Tab 3; our 5–10x speedup is under the same model capacity. **3.3 (inference times of other baseline methods):** In Tab 3, we compared the efficiency with the SA algorithm[20]. For other baselines, we either quoted results from previous work or tested published output[Appendex C]; thus, we don't have the inference time. However, our efficiency is comparable to any conditional text generator (when model capacity is controlled), because we essentially distill the knowledge of searching into a conditional text generator. ♦While our training efficiency is roughly $2\times$(SA+finetuneGPT2), we do not view it as a disadvantage. First, training is usually done off-line; when trained, our model is very efficient for deployment. Second, it's understandable that we sacrifice some training efficiency compared with supervised models, since we do not have parallel data. In fact, our approach should be more efficient (and labor-saving) than data collection plus human annotation in the supervised setting, as explained in "Broader Impact." We hope our explanation could address the review's concern of efficiency. We will discuss this in the revision.

**R5: 3.1:** Thanks for understanding the significant substance of our work! See response to 5.1–5.4. **3.2 (search-based):** Yes, simulated annealing can be thought of as stochastic search, because it aims to maximize a function. We'll explain more. **3.3/3.4:** Ablation study is in Tab 3, showing SA performance within TGSL. We briefly discussed StructuredPrediction (and RL) in §2.4: StructPredict in the supervised setting requires expert demonstrations, but we only have pseudo-reference given by SA. We'll include the suggested papers and discuss StructPredict more in the revision. **4.1:** Thanks for pointing this out. Yes, we've already realized this typo and corrected it in our local version. **4.2:** Energy $E(x) \in (-\infty, \infty)$ is indeed unbounded and unnormalized. Probability is thus defined as $p(x) \propto \exp\{-E(x)\}$. We'll present it when revising the paper. **5.1:** $\Delta=1$ like most applications. **5.2:** Epoch=6[Line233] by early stop on validation. **5.3:** Beam size=5[Line 285]. **5.4:** Yes. $N$ is the length of y for a sample. Thanks! Will clarify.



[Meta-Review · NeurIPS 2020]

The reviewers overall found the method here interesting, but there were a few concerns: 1. Most importantly, the description in the paper seems to overclaim the breadth of the results. For example given the relatively limited scope of the experiments it seems strange to call this "Unsupervised Text Generation", at the best it's probably "Unsupervised Paraphrasing", "Unsupervised Style Transfer", or stretching "Unsupervised Conditional Text Generation". I would encourage the authors to scope the claims appropriately. 2. Significant additional complexity compared to other methods. The method is relatively complicated, and this may limit its applicability to some extent. However, despite these concerns I do think that the paper is of sufficient quality to be published at NeurIPS and thus recommend it for acceptance.